# Examining the Negative Sentiments Related to Influenza Vaccination from 2017 to 2022: An Unsupervised Deep Learning Analysis of 261,613 Twitter Posts

**DOI:** 10.3390/vaccines11061018

**Published:** 2023-05-23

**Authors:** Qin Xiang Ng, Dawn Yi Xin Lee, Clara Xinyi Ng, Chun En Yau, Yu Liang Lim, Tau Ming Liew

**Affiliations:** 1Health Services Research Unit, Singapore General Hospital, Singapore 169608, Singapore; ng.qin.xiang@u.nus.edu; 2MOH Holdings Pte Ltd., 1 Maritime Square, Singapore 099253, Singapore; 3School of Medicine, Dentistry and Nursing, University of Glasgow, Glasgow G12 8QQ, UK; 4NUS Yong Loo Lin School of Medicine, Singapore 117597, Singapore; 5Department of Psychiatry, Singapore General Hospital, Singapore 169608, Singapore; 6SingHealth Duke-NUS Medicine Academic Clinical Programme, Duke-NUS Medical School, Singapore 169857, Singapore; 7Saw Swee Hock School of Public Health, National University of Singapore, Singapore 117549, Singapore

**Keywords:** influenza, flu, vaccine, hesitancy, negative sentiment, machine learning

## Abstract

Several countries are witnessing significant increases in influenza cases and severity. Despite the availability, effectiveness and safety of influenza vaccination, vaccination coverage remains suboptimal globally. In this study, we examined the prevailing negative sentiments related to influenza vaccination via a deep learning analysis of public Twitter posts over the past five years. We extracted original tweets containing the terms ‘flu jab’, ‘#flujab’, ‘flu vaccine’, ‘#fluvaccine’, ‘influenza vaccine’, ‘#influenzavaccine’, ‘influenza jab’, or ‘#influenzajab’, and posted in English from 1 January 2017 to 1 November 2022. We then identified tweets with negative sentiment from individuals, and this was followed by topic modelling using machine learning models and qualitative thematic analysis performed independently by the study investigators. A total of 261,613 tweets were analyzed. Topic modelling and thematic analysis produced five topics grouped under two major themes: (1) criticisms of governmental policies related to influenza vaccination and (2) misinformation related to influenza vaccination. A significant majority of the tweets were centered around perceived influenza vaccine mandates or coercion to vaccinate. Our analysis of temporal trends also showed an increase in the prevalence of negative sentiments related to influenza vaccination from the year 2020 onwards, which possibly coincides with misinformation related to COVID-19 policies and vaccination. There was a typology of misperceptions and misinformation underlying the negative sentiments related to influenza vaccination. Public health communications should be mindful of these findings.

## 1. Introduction

The individual and public health benefits of influenza vaccination cannot be overemphasized. Influenza is a global health threat, with outbreaks accounting for 3 to 5 million cases of severe illness and approximately 250,000 deaths every year [1,2]. On top of that, it poses a huge economic burden and contributes to higher medical costs, with the estimated economic burden of seasonal influenza in the United States more than USD 11 billion in 2015 [3]. While not perfect, receiving an up-to-date influenza vaccination against the circulating strains is able to effectively reduce an individual’s infection, transmission, and disease severity risk [4].

During the Coronavirus Disease 2019 (COVID-19) pandemic, most countries reported low circulation of influenza and other respiratory viruses, with the implementation of public health measures such as social distancing [5]. However, with the relaxation of these restrictions, many countries are now seeing surging numbers of influenza cases as well as more patients requiring critical care. Influenza cases will likely continue to rise as more countries lift COVID-19 movement, masking, and social restrictions [6], making the need to receive an influenza vaccine an even more pressing one.

Despite the availability of the vaccines, extensive research and reassuring public safety data [7,8], there appears to be generally low influenza vaccination uptake over the years. Influenza vaccination coverage ranges from 2.4% in 2014 [9] to 9.4% in 2018 in China [10], 41.5% in 2007 to 30.8% in 2018 in Republic of Korea [11], 37.1% in 2018 [12] to 31.8% in 2022 in Saudi Arabia [13], and 40.4% in 2010 to 49.4% in 2021 in the United States [14]. Even more worryingly, multiple studies that examined the public sentiments surrounding Coronavirus Disease 2019 (COVID-19) vaccination have found new heights of vaccine hesitancy or refusal [15,16], and hints of possible negative spill-over effects to influenza and other childhood immunizations [17]. 

In the recent COVID-19 pandemic, social media has contributed to the proliferation of misinformation [18], undermining public trust and contributing to vaccine hesitancy amongst the public [19]. Furthermore, it is concerning that influential figures, particularly right wing politicians, utilize platforms such as Twitter for political discourse to shape public perception on pandemics [20,21]. With more than 200 million daily active users [22], Twitter is a powerful platform for the rapid dissemination of information and expression of emotions. Given the wealth of information on Twitter, it has served to be useful in assessing public views on health issues [23,24]. Sentiment analysis of tweets can thus help to better understand public perception toward vaccination. A recent study using sentiment analysis has demonstrated that tweet sentiment score may help to predict vaccination rates, with negative sentiment being predictive of lower vaccination rates [25]. Given that misinformation tended to be emotionally driven, demonstrate more negative sentiments such as fear and anger [26,27], and that anti-vaccine Twitter users were more likely to spread unreliable information [28], we chose to focus on negative sentiment analysis to provide insight into reasons for resistance to vaccination and to identify potential sources of misinformation. 

The purpose of our study was to investigate the prevailing negative sentiments related to influenza vaccination by analyzing public Twitter posts from 2017 to 2022 using deep learning techniques. As vaccine acceptance rates can change over time [29], the present study will also look at the temporal trends related to public concerns about influenza vaccination. In particular, this study sought to examine whether there has been any change in the temporal trends in the past two to three years vis à vis the ongoing COVID-19 pandemic and to characterize these negative sentiments. The results of our study could be useful for public health practitioners and policy makers in understanding the public’s negative perceptions toward the vaccine and developing strategies to address them.

## 2. Methods

The methodology for the present study was adapted from previous infodemiology studies, which substantiated the use case for machine learning approaches to analyze unstructured free text over Twitter as a novel method to explore public sentiments and understand emotional manifestations on a given topic [15,30]. Tweets posted in the English language between 1 January 2017 and 1 November 2022 were collected, with retweets and duplicate tweets excluded. The origin of the tweets was not limited to any specific country, as long as they were in English.

For natural language processing (NLP), Bidirectional Encoder Representations from Transformers (BERT), a state-of-the-art deep machine learning approach, was used. BERT utilizes unsupervised masked language model (MLM) and unsupervised next sentence prediction (NSP) for text deep pre-training and fine-tuning using publicly available two-class sentiment datasets, as opposed to the traditional bag-of-words model [31]. The BERT-based model has the added advantage of requiring less text pre-processing prior to analysis since it provides sentence context.

Named Entity Recognition was then used to identify individual users, focusing on location, organizations, person, and miscellaneous entities [32]. To accomplish this, actual human names were used to identify the Twitter account associated with each post. Finally, sentiment analysis was performed using the pre-trained SieBERT model, which classifies text as either positive or negative sentiment, with an accuracy of approximately 93% demonstrated by its use on various datasets [33]. We specifically focused on analysis of negative sentiments as this may spread faster and be more influential to individuals compared to positive sentiment tweets [34]. Topic modelling, specifically BERTopic, was employed to generate interpretable key concerns [35] on the prevailing negative sentiments related to influenza vaccination. We have included sample codes used in the methodology and additional sample tweets for each topic in the Appendix A. 

Once the required data were gathered, the researchers conducted thematic analysis using an iterative and inductive approach as described by Braun and Clarke [36]. Initially, the investigators independently reviewed the topic labels and themes and acquainted themselves with the sample tweets and relevant keywords. They then created preliminary codes and established overall themes, which were subsequently reviewed and refined. The themes were defined and specified, and a write-up was produced. The research team held regular meetings to discuss the emerging themes and resolved any coding discrepancies through team discussion until a consensus was reached.

Ethics approval for the study was granted by the SingHealth Centralised Institutional Review Board of Singapore (reference number: 2021/2717). This study did not directly involve human participants. All data used in the present study were collected in accordance with Twitter’s terms of use.

## 3. Results

### 3.1. Search Results

A total of 1,152,181 initial tweets were identified in the period of 1 January 2017–1 November 2022. After removing duplicate tweets, tweets by organizations or news outlets, and tweets without relevant terms—‘flu jab’, ‘#flujab’, ‘flu vaccine’, ‘#fluvaccine’, ‘influenza vaccine’, ‘#influenzavaccine’, ‘influenza jab’, or ‘#influenzajab’—451,923 tweets remained. Of these, 261,613 tweets were identified to be negative sentiment tweets. The flowchart showing the tweet selection process is displayed in Figure 1.

### 3.2. Topic Modelling

A total of five topics were generated via BERTopic. The total prevalence of these five topics was 88.9%; the remaining 11.1% was from a topic that was omitted from the current results as the BERT NLP model generates a Miscellaneous topic that groups all remaining (unfitted) tweets together. These unfitted tweets were classified as outliers based on the NLP model. This also helped reduce noise that can affect the topic representations.

A significant majority of the tweets were centered around Topic 1 (*n* = 221,732 tweets, 84.8%), containing criticisms regarding perceived influenza vaccine mandates or coercion to vaccinate. Topics were numbered 1 to 5 in descending order based on their prevalence. The topics were grouped into two main themes by qualitative thematic analysis, namely criticisms of governmental policies related to influenza vaccination and misinformation related to influenza vaccination. Table 1 contains the details of the topics within each theme.

### 3.3. Analysis of Temporal Trends

To examine the temporal trends of tweets in the two identified themes and individual topics, we analyzed the prevalence of tweets as a function of the number posted over the past five years (Figure 2 and Figure 3). Our analysis revealed a notable increase in prevalence of tweets across all themes and topics from year 2020 onwards, with a particularly sharp increase in frequency of tweets related to Topic 2, which possibly coincides with the onset of the COVID-19 pandemic. Notably, there was also a significant increase in frequency of tweets regarding Topic 4 in year 2020 as well, corresponding with the early stages of COVID-19 vaccinations development and initiatives.

In addition, we analyzed the sentiment across all tweets and the changes over the past five years by plotting the proportion of negative tweets as a function of the total number of tweets (Figure 4). Prior to year 2020, the proportion of negative sentiments escalated toward and through Fall each year but was not dominant for a prolonged period. However, from year 2020, coinciding with the onset of the COVID-19 pandemic, negative tweets rose significantly and persisted thereafter.

### 3.4. Analysis of Geolocational Data

We pulled the geolocation for 124,418 of the unique tweets identified in our study, with each unique tweet being indicated as a black dot in the map (Figure 5). Our data revealed that a large majority of the users tweeting originated from Europe (*n* = 34,938, 28.1%) and North America (*n* = 33,626, 27.0%). The remaining tweets were distributed across the globe, and originated from Africa (*n* = 1294, 1.04%), Asia (*n* = 2010, 1.62%), Australia (*n* = 3885, 3.12%) and South America (*n* = 530, 0.43%). However, we were unable to confirm the location of origin for a portion (*n* = 48,135, 38.7%) of unique tweets due to privacy and data restrictions on Twitter elaborated under our discussion.

## 4. Discussion

In this infodemiology study, we utilized unsupervised deep learning to analyze a corpus of free text from social media tweets containing negative sentiments related to influenza vaccination. Our analysis revealed a typology of beliefs, misperceptions and misinformation underlying the negative sentiments related to influenza vaccination. In particular, the significantly polarizing nature of public opinions toward COVID-19 vaccination may have adversely affected influenza vaccination sentiments.

Our analysis found a clear shift in trends of sentiments toward influenza vaccination since the onset of COVID-19. Prior to the pandemic, there was no dominant discourse on Twitter, and tweet frequency across all analyzed topics was significantly lower. However, the frequency of tweets increased sharply in year 2020 following the onset of the COVID-19 pandemic, which may reflect the unprecedented impact of the COVID-19 pandemic on public discourse, heightened public awareness and interest in vaccination-related topics. Negative sentiments have also consistently dominated the discourse on Twitter since then, which possibly coincides with misinformation related to COVID-19 policies and vaccination. Our findings are not surprising, given that social media platforms have been found to contribute to the amplification of vaccine questioning and vaccine hesitancy [17]. Additionally, there appears to be some transference of negative thoughts and feelings to influenza vaccination, as can be seen in Topics 3, 4 and 5 which contained skepticism regarding COVID-related policies, namely the effectiveness of masks to prevent respiratory infections, priority groups to receive the COVID vaccine and concerns with the use of mRNA in vaccines. Our findings align with a recent study in the United States which found a decrease in adult influenza vaccine uptake following the availability of COVID-19 vaccines in the 2012–2022 season, particularly in states at the bottom two quartiles of COVID-19 vaccine uptake. Given the polarizing nature of vaccination against COVID-19, it is hypothesized that perceptions toward COVID-19 and COVID-19 vaccines may have spilled over and influenced attitudes toward influenza vaccines [37]. Another study conducted in the United States also found substantial overlap and potential spillover of vaccine hesitancy across vaccines over the course of the pandemic [38]. This could possibly be a result of the mental association between COVID-19 and influenza [39], which may be attributed to the similarities in symptoms and mode of transmission between the viruses [40]. 

Our findings corroborate a recent report which examined nationally representative data from the United States Centers for Disease Control and Prevention (CDC) and found low COVID-19 vaccination rates to be associated with low influenza vaccination rates [37]. Another study in Greece also found that intention to receive a COVID-19 vaccine and previous influenza vaccination behavior are strong predictors of influenza vaccine willingness [41]. Similarly, another study found that individuals who received the influenza vaccine in previous years were also more likely to receive the COVID-19 vaccine [42]. While vaccine uptake rates may vary across regions and populations, these studies indicate a possible relationship between the two vaccines. Although correlation does not necessarily imply causation, this relationship certainly warrants further study. It is often assumed that misinformation and false beliefs about vaccines can be easily corrected but studies have learned that misinformation can have a continued influence effect on people’s thinking even after they become aware of a correction and accept it to be true [43,44]. This is exemplified through Topic 5, which showed perpetuation of false beliefs that the influenza vaccine also employs a mRNA vaccine technology platform similar to COVID-19 vaccines. One’s held beliefs about the COVID-19 vaccines may also affect their willingness to vaccinate against the flu and vice versa. 

While previous studies have found that credibility of correction source matters [45], addressing the issue of vaccine hesitancy is not a straightforward task given the highly polarized nature of vaccine conversations [46]. A study on over 54 million users found that users interacting with conspiracy topics tended to exist in their own well-formed and highly segregated echo chambers, ignoring dissenting information [47]. This suggests that debunking may not be as effective in addressing anti-vaccine sentiments, given the limited interaction of conspiracy users with these debunking posts [47]. However, new strategies such as reactive debunking and particularly preemptive information distortion strategies offers significant promise [48,49,50,51,52,53]. Alternative measures such as tighter social and mass media regulation can be considered especially in the context of misinformation. Given that anti-vaxxers and partisan actors have been found to be the most influential in shaping anti-vaccine sentiments on Twitter [46], it is thus important to regulate the content posted by this small but influential group of users. This has proven to demonstrate potential in the reduction in misinformation in other contexts, for example, the de-platforming of Trump resulted in at least a temporary reduction in misleading information about election fraud [54]. Further research is required to develop more effective and comprehensive strategies to combat vaccine hesitancy and misinformation.

In terms of health messaging, the proportion of negative sentiments appeared to escalate towards and through the Fall of each year, as seen in Figure 4, which (at least in the United States) is usually the time when health authorities tend to roll out their flu vaccination campaigns for each year. The fact that this seesaw pattern increased after 2020 suggests some influence of the pandemic and the vaccine program on this pattern. This has potential implications for how health authorities might consider the timing of vaccination campaigns, and authorities may consider coordinating primary vaccination all at the same time. However, the findings may also be historically idiosyncratic to this particular pandemic time. There is a noticeable decrease in mask-related tweets after 2021 as most countries have now lifted masking mandates in public spaces [55].

Based on prior studies, it has been identified that there are several reasons why people may be hesitant to receive the influenza vaccine, and these reasons can vary depending on a multitude of factors, such as personal beliefs, experiences, and access to health care [41,56]. Some individuals may have concerns or hold misconceptions about the safety or efficacy of the vaccine. General vaccine misinformation also predicted vaccination hesitancy. A previous study performed in the United States using a nationally representative sample found that a significant percentage of the public (up to 43%) believed that the flu vaccine itself can cause the flu in recipients [57]. It is relevant to understand the prevailing negative sentiments toward influenza vaccination, as similar to COVID-19 vaccines, the flu virus changes every year, and yearly vaccination is recommended as the flu jab is formulated each year to protect against the strains of the virus that are expected to be most common during that flu season [58]. Influenza is a highly contagious respiratory illness caused by the influenza virus, and it can cause serious complications, particularly in vulnerable populations such as the elderly, young children, and those with multimorbidity and weakened immune systems [59]. 

Critically, it is important to note that in our study, the findings appear to suggest that more people were hesitant to receive the flu vaccine (or any vaccine) due to a general mistrust of the government’s health policies and the health care system, rather than concerns about the safety and efficacy of vaccines per se. This could be driven by the proliferation of misinformation and “fake news” over online platforms; misinformation can significantly affect one’s receptivity to the flu vaccine or any vaccine in general as it creates confusion, fear, and distrust among people [60]. Previous studies have cited political and media misinformation as a driver of distrust of authorities [60,61]. Increases in conspiracy beliefs were associated with decreases in one’s belief in and support for COVID-19 public health recommendations and preventive measures [61]. Vaccine hesitancy is a deeply complex and emotionally charged issue and more must be done to investigate effective ways to address misinformation. It is often thought that providing accurate information about the influenza vaccines would automatically help people make informed decisions about their health, unfortunately, correcting myths about flu vaccines may not be effective in prompting one’s behavioral intent to get vaccinated [57]. Vaccination hesitancy can still persist and is difficult to change. Health policy leaders should formulate a multi-faceted approach that involves providing accurate information, addressing misconceptions, and engaging in dialogue with those who hold differing views. 

Our study is not without limitations. Firstly, we aimed to explore the phenomenon of negative sentiments on Twitter in an exploratory manner. As such, we did not have any pre-established theoretical framework guiding our research. While the absence of a theoretical framework allowed us to approach the topic with an open mind, and explore various aspects, patterns, and relationships that emerged during this study, we acknowledge the lack of theoretical grounding for our study and the potential lack of generalizability of the typology uncovered herein. Secondly, while we tried to capture a range of tweets with our keywords, there is a possibility a proportion of anti-vaccination tweets may not contain these specific words. For instance, tweets with hashtags such as #BigPharma and #StopMandatoryVaccines may convey negative attitudes towards influenza vaccination without directly referencing the keywords. As a result, our search may not have accommodated such alternate topics, which limits the scope of our findings. Thirdly, the analysis was based on Twitter posts and only tweets in English were eligible for inclusion, hence the findings may not be generalizable to non-Western countries and communities as the bulk of Twitter users are thought to be from North America, leaving Asian countries to be underrepresented [62]. Due to the restriction of Twitter use since 2009, China in particular is underrepresented in our analysis [63]. It is also important to note that there are other prominent social media platforms aside from Twitter, that could be examined in future research to allow for a more comprehensive understanding of public sentiment toward influenza vaccination. Moreover, while we were able to pull some geolocational data for the unique users we identified in our study, we were unable to fully do so as we lack details on the country of origin of the tweets and demographics of the posters due to privacy and data restrictions. Twitter removed the option to share the geolocation and coordinates of tweets in 2019 [64]. Even before this, only an exceedingly small portion of Twitter users would post their coordinates, and they are not representative of the wider population of Twitter users [65]. Fourthly, in our study, we utilized sentiment analysis based on machine learning which comes with its own strengths and limitations. While it allows for the study of a large number of tweets otherwise not possible with human coding, we acknowledge that it may not be as accurate as the latter, especially with unsupervised machine learning. Additionally, as the coding was performed in team through an iterative matter via discussion and consensus, we were not able to calculate intercoder reliability. Furthermore, the pre-trained SieBERT model we utilized for sentiment analysis is only capable of classifying tweets into negative versus positive sentiment [32], thus we were unable to differentiate amongst the different types of negative sentiments, e.g., fear, anger and shame. This is a significant drawback as research has demonstrated that sadness and anger tend to diffuse differentially [66]. Fifthly, in our analyses, we also focused solely on negative sentiments and combined positive and neutral/unclassifiable into the same category, which may obscure certain nuances and distinctions. Moreover, some tweets that appear affect neutral or positive may convey negative beliefs. Despite the concerns about the potential limitations in terms of accuracy of automated sentiment analysis compared to human interpretation, SieBERT has demonstrated high accuracy rates of approximately 93% when tested on various datasets [32], giving us confidence in its ability to accurately identify tone of datasets of text. Sixthly, the tweets collected for analysis were not necessarily from unique users as we cannot exclude the possibility that a small number of active users may tweet multiple times on the same subject and bear overlapping sentiments. This may affect our results and interpretations. Lastly, despite our efforts to include tweets by users with real human names and excluding retweets and duplicated tweets, we were unable to completely eliminate nonhuman Twitter users (i.e., bots) posing as genuine users from our study sample. Given that it is unclear to what extent artificial intelligence is transforming the variability of bot-based tweets, we acknowledge that this measure alone may not be sufficient guardrail. A recent study on social bot’s involvement in the COVID-19 vaccine discussions found that 8.87% of the users identified were bots, and these bots contributed to 11% of the tweets [67]. These bots may have been designed to manipulate public opinion and disseminate false information, which could potentially impact our analyses. Nonetheless, the tweets put out by these bots are in the digital circulatory system and their expressed sentiment matters as it is part of what people are (potentially) exposed to.

## 5. Conclusions

In conclusion, the present infodemiology study found a typology of beliefs, misperceptions and misinformation underlying the negative sentiments related to influenza vaccination, consistent with previous studies. This appeared to have increased in prevalence from the year 2020 onwards, which coincides with misinformation related to COVID-19 policies and vaccination. The potential “spillover” effects and the negative impact of vaccine misinformation is significant and cannot be understated. The COVID-19 pandemic may have corollary impacts on influenza vaccine uptake. On one hand, the pandemic may have raised awareness about the importance of vaccines and led to increased funding for vaccine research and development. On the other hand, the pandemic may have fueled vaccine hesitancy and misinformation globally. Future study should further investigate specific policies and messaging to combat misinformation and promote influenza vaccination to the public.

## Figures and Tables

**Figure 1 vaccines-11-01018-f001:**
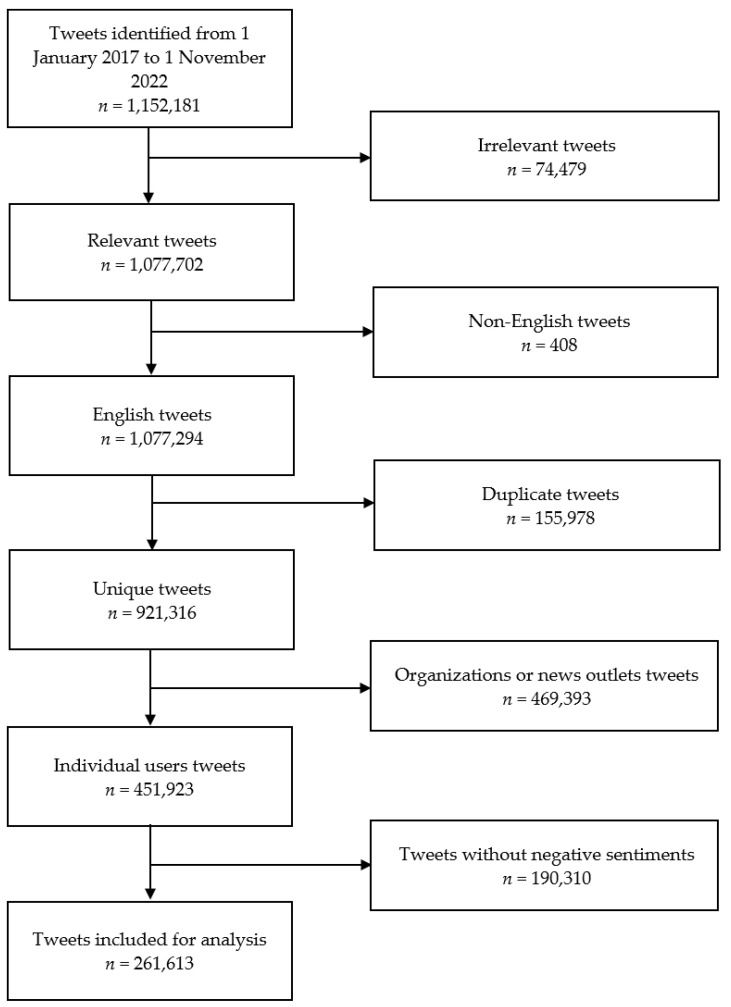
Flowchart illustrating the tweet filtering and selection process.

**Figure 2 vaccines-11-01018-f002:**
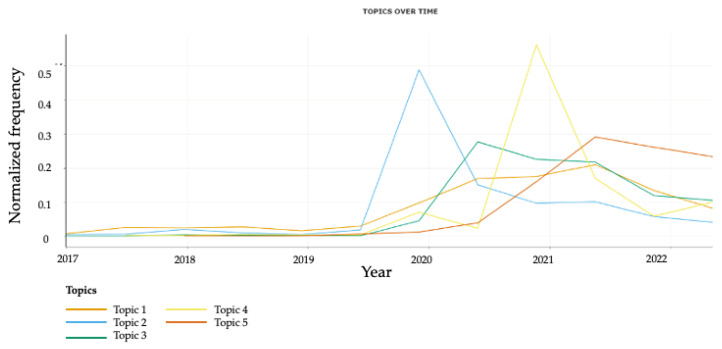
Temporal trends in normalized frequency of tweets in each individual topic.

**Figure 3 vaccines-11-01018-f003:**
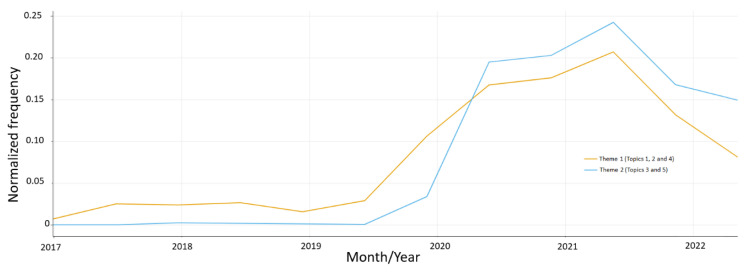
Temporal trends in the normalized frequency of tweets belonging to Theme 1 (Topics 1, 2 and 4) and Theme 2 (Topics 3 and 5).

**Figure 4 vaccines-11-01018-f004:**
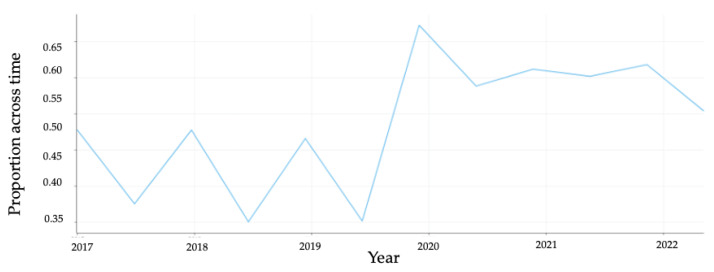
Temporal trends for negative sentiment tweets in terms of proportion over time.

**Figure 5 vaccines-11-01018-f005:**
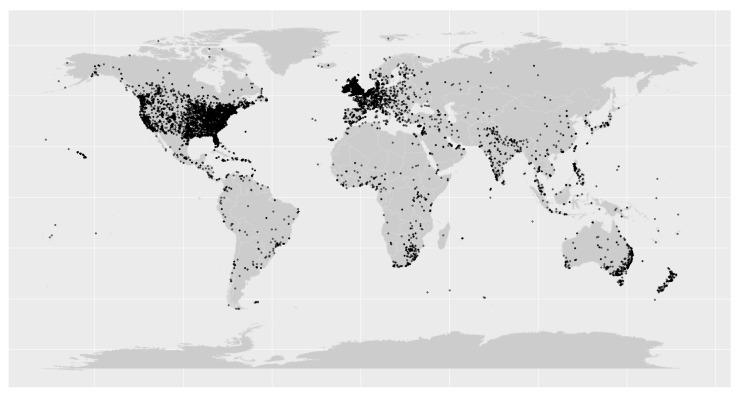
Geolocations of the individual tweets included in this study.

**Table 1 vaccines-11-01018-t001:** Themes related to the negative public sentiments toward influenza vaccination, along with the respective topics and sample tweets (N = 261,613).

Theme and Topic (*Keywords*)	Sample Tweets	Number of Tweets, *n* (%)
**Theme 1: Criticism of governmental policies related to influenza vaccination**		
Topic 1: Criticism of the government policy on mandatory influenza vaccine (*effective*, *such as flu vaccine*, *arm*, *GP*, *day*, *NHS*, *strain*, *flu*, *flu shot*)	“What is your problem, what happened to free will? If some people don t want it then why should we force them? Not everyone gets the flu jab every year” “Just had breaking news here in my state where ALL children are going to be required to have the ‘flu vaccine’ before heading back to school. People are pissed as they should be. NO ONE is going to make me get any flu shot if I don’t want one ever. I’ll move to another country	221,732 (84.8)
Topic 2: Criticism on public messaging around influenza vaccination (*trump*, *vaccine work*, *flu vaccine work*, *president*, *guy*, *thinks*, *solid*, *solid flu vaccine*, *solid flu*, *corona*)	“Heard him say that at the rally, but heard a radio interview from last week where he said he didn’t. Never takes flu vaccine either.” “My shock is from when he says he never had flu vaccine, hence I ask if as a politician he never traveled to countries where they are mandatory. I was not focused on this one when I made comment”	5272 (2.0)
Topic 4: Criticism of the government policy on priority groups to receive COVID-19 and influenza vaccine (*asthma*, *asthmatics*, *group*, *asthmatics at risk*, *jab list*, *flu jab list*, *priority*, *JCVI*, *eligible*, *asthmatic*)	“On BBC breakfast this morning they’ve said everyone who has an annual flu jab will get a COVID-19 booster, and these people are clinically extremely vulnerable. Except many asthmatics who get an annual flu jab, including myself, don’t qualify for a COVID-19 vaccine yet….” “What you say is not happening in reality. My husband is asthmatic, a key worker and at 46 has been told he cannot have COVID-19 vaccination as not in priority group. Why is he offered a flu jab annually but not a COVID-19 vaccination as a dangerous respiratory virus?”	1636 (0.6)
**Theme 2: Misinformation related to influenza vaccination**		
Topic 3: Misconception that mask wearing can replace influenza vaccine (*mask*, *masks*, *wear*, *wearing*, *wear mask*, *wearing masks*, *wearing mask*, *mask flu*, *wear masks*, *flu season*)	“If masks can save us from COVID-19, they should be able to save us from flu. So why would anyone want a flu jab considering masks are mandatory??” “I am not bothering with the flu vaccine the masks killed it last year.”	2618 (1.0)
Topic 5: Concerns about mRNA vaccine technology (*MRNA*, *MRNA flu*, *MRNA flu vaccine*, *vaccine MRNA*, *MRNA vaccines*, *technology*, *MRNA vaccine*, *flu vaccine MRNA*, *RNA*, *Moderna*)	“MRNA flu vaccine in the works now as well. Roll on sudden deaths” “So sad…. and criminal! Apparently, the flu jab is now mRNA.”	1374 (0.5)

## Data Availability

The datasets generated during and/or analyzed during the current study are available from the corresponding author on reasonable request.

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
