# Peer review of "Examining the Negative Sentiments Related to Influenza Vaccination from 2017 to 2022: An Unsupervised Deep Learning Analysis of 261,613 Twitter Posts"

_vaccines, 2023, doi:10.3390/vaccines11061018_

Round 1
Reviewer 1 Report (New Reviewer)
Thank you very much for giving me the chance to review this very interesting article, dealing with a crucial issue: negative perception of vaccines on social network.
In this study, authors examine the prevailing negative sentiments related to influenza vaccination via a deep learning analysis of public Twitter posts over the past five years. A total of 26 261,613 tweets were analyzed through topic modelling and thematic analysis, that disclosed five topics grouped under two major themes, (1) criticisms of governmental policies related to influenza vaccination and (2) misinformation related to influenza vaccination. A significant majority of the tweets were centered around perceived influenza vaccine mandates or coercion to vaccinate. The analysis also showed an increase in the prevalence of negative sentiments related to influenza vaccination from the year 2020 onwards, with possible coincides with misinformation related to COVID-19 policies and vaccination.
The article is well organized, well written, and methodologically sound, thus it represents a necessary contribution to the stream of literature looking at health misinformation, in particular on Twitter.
However, at this stage, this work presents one major flaw that I the authors should fix before publication.
The major flaw is the lack of a theoretical grounding and literature review to properly ground the study.
The Introduction goes quickly into the topic not allowing the reader to situate himself/herself in the topic.
The authors only discuss previous literature superficially, only quoting few studies seemingly ignoring the wide literature about Twitter and its relation with health communication, together with the already vast literature about Twitter and Covid-19.
It is therefore seminal to ground the study by introducing the role of Twitter in health information, discussing the results of studies about Twitter and Covid.
There is an extensive literature on these topics.
Most importantly it is fundamental to take into consideration how political institutions/authorities have used Twitter during Covid and how this has led to misunderstanding and misinformation.
These two reference should be considered:
https://doi.org/10.3390/socsci10080294
https://doi.org/10.1093/pubmed/fdaa049
By adding the literature review not only the authotrs will enrich the study but it will allow them to justify their choice to only look at negative sentiments that otherwise seems unjustified.
3) CONCLUSIONS
Conclusions are insufficient. Once the authors have expanded the literature review in the conclusion they should 1) put their results into dialogue with previous/similar studies 2) stess out the contribution of this study 3) clearly express the limitation 4) suggest future research
GOOD LUCK!
Author Response
We thank the reviewer for the insightful comment.
We have updated our introduction to include some the discussion of the role of twitter in health communication and incorporated the two suggested references in our manuscript. “In the recent COVID-19 pandemic, social media has contributed to the proliferation of misinformation [18], undermining public trust and contributing to vaccine hesitancy amongst the public [19]. A recent study on public attitudes toward vaccination found a 24% shift toward negative sentiment on social media in wake of the COVID-19 pandemic, citing concerns about safety, efficacy and side effects of vaccination [20]. Furthermore, it is concerning that influential figures, particularly right wing politicians, utilise platforms such as Twitter for political discourse to shape public perception on pandemics [21,22]. With more than 200 million daily active users [23], Twitter is a powerful platform for the rapid dissemination of information and expression of emotions. Given the wealth of information on Twitter, it has served to be useful in assessing public views on health issues [24,25]. Sentiment analysis of tweets can thus help to better understand public perception toward vaccination. Given that misinformation tended to be emotionally driven, demonstrate more negative sentiments such as fear and anger [26,27], and that anti-vaccine twitter users were more likely to spread unreliable information [28], we chose to focus on negative sentiment analysis to provide insight into reasons for resistance to vaccination and to identify potential sources of misinformation."
Reviewer 2 Report (New Reviewer)
See attached

See attached
Author Response
To begin, we would like to thank the reviewer for taking the time to critique and review our manuscript.
|
S/N |
Comment |
Reply |
|
1 |
24: "We then identified tweets with negative sentiment tweets". Redundant. 41-42: "it poses as a huge economic burden" 43: "in the United States was-more than USD" 65-66: "the present study would will also look at" 187: "there appears to be there appears to be" 201: "Our findings collaborate with a recent report". Findings can corroborate a recent report, coincide with a recent report, reinforce a recent report, but not collaborate with a recent report. 177: "towards": I've never understood the function of the 's' on this word. I suspect it is a European vs. American thing. See also 195. I would not the same for "onwards" 285. 262: "Alas" is relatively anachronistic. 285: Does "with particular coincides" intend to read "which in particular coincides"? 287: Unclear referent for "its": (1) negative sentiment, (2) misinformation, or (3) negatively-valenced misinformation. |
Thank you for the comment. We have made the necessary edits. |
|
2 |
69-71: A 1-sentence claiming that results "could be useful for public health practitioners" is insufficient rationale. How might such information be useful, over and above current types of information? I know some of this is material for the Discussion section, but there is a clear need to establish a clear need for doing the study in the first place. |
Thank you for the comment. We agree with the reviewer and we have added the following paragraph to provide more context for the choice of our study.
“In the recent COVID-19 pandemic, social media has contributed to the proliferation of misinformation [18], undermining public trust and contributing to vaccine hesitancy amongst the public [19]. A recent study on public attitudes toward vaccination found a 24% shift toward negative sentiment on social media in wake of the COVID-19 pandemic, citing concerns about safety, efficacy and side effects of vaccination [20]. Furthermore, it is concerning that influential figures, particularly right wing politicians, utilise platforms such as Twitter for political discourse to shape public perception on pandemics [21,22]. With more than 200 million daily active users [23], Twitter is a powerful platform for the rapid dissemination of information and expression of emotions. Given the wealth of information on Twitter, it has served to be useful in assessing public views on health issues [24,25]. Sentiment analysis of tweets can thus help to better understand public perception toward vaccination. Given that misinformation tended to be emotionally driven, demonstrate more negative sentiments such as fear and anger [26,27], and that anti-vaccine twitter users were more likely to spread unreliable information [28], we chose to focus on negative sentiment analysis to provide insight into reasons for resistance to vaccination and to identify potential sources of misinformation.“ |
|
3 |
89-90: You should at least explain what goes into SieBERT's construction of "negative sentiments." This is important because, for example, sadness (e.g., over the loss of a loved one) and anger (e.g., at the mask mandate) are potentially quite distinct in their diffusion implications, yet they both represent "negative sentiments." In general, I would expect sadness tweets to decrease, but anger tweets to increase, over the course of the pandemic (for various reasons). Thus, depending on proportions, these kinds of trends might help contextualize the "significant increase in negative sentiment" (151). |
Thank you for the comment. We agree with the suggestion that certain nuances may be missed in our classification, however, due to the inherent limitations of SieBERT, it is only able to it predict either positive (1) or negative (0) sentiment. The model utilised in the study has been finetuned and evaluated on 15 data sets from diverse text sources.
Nonetheless, we acknowledge this limitation and have added this to our discussion of study limitations.
“Furthermore, the pre-trained SieBERT model we utilized for sentiment analysis is only able to predict either positive (1) or negative (0) sentiment [32], thus we were unable to differentiate amongst the different types of negative sentiments, e.g. fear, anger and shame.” |
|
4 |
Fig. 4: It is suspicious that the non-negative and negative lines are exact mirror images. I'm assuming that this is because "non-negative sentiment" is all remaining content when "negative sentiments" are subtracted, which makes these graph lines functionally redundant, and thus, the non-negative line is redundant (and confusing). Fig. 4: The trend lines up to and through 2020 suggest the possibility that negative sentiments escalate toward and through the Fall of each year, which (at least in the U.S.) is when health authorities tend to roll out their flu vaccination campaigns and the new vaccine type is released for each year. The fact that this seesaw pattern implodes after 2020 suggests the interference of the pandemic and the vaccine program with this pattern. This would seem to have potential implications for how health authorities might consider the timing of vaccination campaigns. Is it better to try and coordinate primary vaccination all at the same time, or to offset (e.g., COVID 19 in spring, and flu in the fall)? |
Thank you for the comment. We apologise for the confusion caused and we would like to clarify that the labels were meant to be “proportion” rather than “prevalence”. We have edited our results segment and discussion accordingly. “Prior to year 2020, proportion of negative sentiments escalated toward and through Fall each year, but was not dominant for a prolonged period of time. However, from year 2020, coinciding with the onset of the COVID-19 pandemic, a significant increase in proportion of negative sentiment and decrease in proportion of non-negative sentiment was observed. Since then, negative sentiment has been predominant.”
|
|
5 |
90-91: That "negative sentiments ... may spread faster..." is a weak justification for ignoring positive sentiments. The evidence on virality (which can include more dimensions than velocity; e.g., acceleration, popularity, reach or duration/persistence, fecundity, etc.) based on sentiment is actually somewhat mixed in the context of disease outbreaks. Health crises and pandemics are likely by their nature to trend at least partially negative in memetic and topical content (Boukobza et al., 2022; Kaminski et al., 2021; Krawczyk et al., 2021; Wicke & Bolognesi, 2021), and transparency in negative health messaging may be required to maintain societal trust in such situations (Petersen et al., 2021). |
We thank you for your insightful comment and the references provided. We have added the following statement to justify our choice of using negative sentiment for analysis. “Given that misinformation tended to be emotionally driven, demonstrate more negative sentiments such as fear and anger [26,27], and that anti-vaccine twitter users were more likely to spread unreliable information [28], we chose to focus on negative sentiment analysis to provide insight into reasons for resistance to vaccination and to identify potential sources of misinformation. ”
|
|
6 |
124-125: "by qualitative thematic analysis" is extremely vague. Was any intercoder reliability assessed? Were any standardized qualitative approaches applied? |
We thank you for the comment. Intercoder reliability was not assessed as the analysis was based on consensus between the researchers. Given the nature of the dataset, it does not strictly fit into any of the QUAL research traditions. Nonetheless, the analysis was performed according to the methods outlined by Braun and Clarke (2006). Once the required data was gathered, the researchers conducted thematic analysis using an iterative and inductive approach as described by Braun and Clarke (2006). Initially, the investigators independently reviewed the topic labels and themes and acquainted themselves with the sample tweets and relevant keywords. They then created preliminary codes and established overall themes, which were subsequently reviewed and refined. The themes were defined and specified, and a write-up was produced. The research team held regular meetings to discuss the emerging themes and resolved any coding discrepancies through team discussion until a consensus was reached. |
|
7 |
Table 1: Is there a conceptual reason why the topic enumeration is out of order? 1, 2, 4, 3, 5?
|
We thank you for the comment. We apologise for the confusion and we have added the following statement to clarify the order of the topics. “Topics were numbered 1 to 5 in descending order based on their prevalence.” |
|
8 |
Table 1: The misinformation topics seem relatively unique to the confluence of COVID-19 with flu, as it seems unlikely these would be major topical themes in studies prior to COVID-19. So, this begs the question whether your results are historically idiosyncratic to this particular pandemic time. This is further suggested by your finding (137-139) about the decrease in mask-related tweets.
|
Thank you for the comment. We agree with the reviewer that our findings may be historically idiosyncratic to the COVID-19 pandemic and have updated our discussion to include the following point. “In terms of health messaging, the proportion of negative sentiments appeared to escalate towards and through the Fall of each year, as seen in Figure 4, which (at least in the US) is usually the time when health authorities tend to roll out their flu vaccination campaigns for each year. The fact that this seesaw pattern increased after 2020 suggests some influence of the pandemic and the vaccine program with this pattern. This has potential implications for how health authorities might consider the timing of vaccination campaigns, and authorities may consider to try and coordinate primary vaccination all at the same time. However, the findings may also be historically idiosyncratic to this particular pandemic time. There is a noticeable decrease in mask-related tweets after 2021 as most countries have now lifted masking mandates in public spaces [55]. This means that the observed trends may change or even reverse with time.” |
|
9 |
34-135: "Our analvsis revealed a notable increase in prevalence of 134 tweets across all themes and topics". Not according to Fig. 2, since only 4 topics are graphed. Where is Topic #5? Obviously it would not occur until at least 2020 or 2021, but if it's one of your 5 topics, why is it not graphically represented in Fig. 2? |
We thank you for the comment. We apologise for the oversight and we have now added Topic 5 into our graph in Figure 2. |
|
10 |
Fig. 5: It is hardly surprising that Asian countries in general, and China in particular, are underrepresented, given that only Twitter (instead of Weibo) and English language are used for selection. It seems some statement should be made to this effect. You only very briefly deal with this in 263-266. |
We thank you for the comment. We agree and we have added this to our discussion of study limitations.
“The analysis was based on Twitter posts and only tweets in English were eligible for inclusion, hence the findings may not be generalizable to non-Western countries and communities as the bulk of Twitter users are thought to be from North America, leaving Asian countries to be underrepresented [62]. Due to the restriction of Twitter use since 2009, China particular is also underrepresented in our analysis [63].” |
|
11 |
173-174: In referring to your results as producing "a typology of beliefs, misperceptions and misinformation underlying the negative sentiments related to influenza vaccination", you seem to imply that this typology has some theoretical and pragmatic generalizability. This is contingent upon a variety of conditions, however, including (1) the preliminary search parameters (i.e., 'fu jab', '#flujab', 'flu vaccine', '#fluvaccine', 'influenza 109 vaccine', "#influenzavaccine', 'influenza jab', or '#influenzajab', ), which are biased toward misinformation and negative sentiment), and (2) the extraordinary diversity of potential search parameters that might be relevant to such a typology. For example, a study of tweets referring to a child vaccine mandate in California included tags such as #BigPharma, #SB277, and #StopMandatoryVaccines. Many tweets that are generically anti-vax would be semantically and pragmatically inclusive of the flu vaccine yet might not use the word "flu" or "influenza" in them. The point is that there is likely a vast domain of negatively-oriented tweets that do not explicitly use the term "flu" in them. I recognize the problem that you would face, given what is no doubt an extraordinarily disproportionate number of COVID-19 vaccine related tweets, but it nevertheless begs the question as to how we are to judge the comprehensiveness or representativeness of your initial search terms. A comprehensive typology needs to accommodate such alternate topics. |
We thank you for the comment. We agree and we have added this to our discussion of study limitations. "Our study is not without limitations. While we tried to capture a range of tweets with our keywords, there is a possibility a proportion of anti-vaccination tweets may not contain these specific words. For instance, tweets with hashtags such as #BigPharma and #StopMandatoryVaccines may convey negative attitudes towards influenza vac-cination without directly referencing the word “flu” or “influenza”. As a result, our search may not have accommodated such alternate topics, which limits the theoretical and pragmatic generalizability of our findings." |
|
12 |
A related concern is the possibility that many tweets that are relatively "affect neutral" might still be negatively valenced in belief or attitude: e.g., "Requiring a vaccine mandate is unconstitutional" or "If Big Pharma thinks it will control me, it has another thing coming." In contrast, potentially positively-valenced statements might also present problems: e.g., "I would personally be happy (overjoyed) to give Dr. Fauci his walking papers." If we don't have any human oversight of such nuances, it is hard to know the extent to which an automatically classified set of topics represents. I would expect to see the entire sample correlated to a gold-standard data set of validated negative sentiment tweets |
We thank you for the comment. We agree and we have added this to our discussion of study limitations.
“Moreover, some tweets that appear affect neutral or positive may convey negative beliefs. Despite the concerns about the potential limitations in terms of accuracy of automated sentiment analysis compared to human interpretation, SieBERT has demonstrated high accuracy rates of around 93% when tested on various datasets [32], giving us confidence in its ability to accurately identify tone of datasets of text.” |
|
13 |
177-178: "a clear shift in trends of sentiments towards influenza vaccination since the onset of COVID-19". It is clear in one sense -there is a significant increase in amount of negative tweets. However, in another sense, it is not clear, in that the previous seesaw pattern has dissolved, and no clear pattern (e.g., increase over time, decrease over time) emerged over the course of the pandemic. This suggests the potential for a peripheral novel pandemic disease and the necessary associated governmental, societal and social disruptions to interfere with patterns of vaccine sentiments. |
Thank you for the comment. We have updated our discussion to include the following point.
“In terms of health messaging, the proportion of negative sentiments appeared to escalate towards and through the Fall of each year, as seen in Figure 4, which (at least in the US) is usually the time when health authorities tend to roll out their flu vaccination campaigns for each year. The fact that this seesaw pattern increased after 2020 suggests some influence of the pandemic and the vaccine program with this pattern. This has potential implications for how health authorities might consider the timing of vaccination campaigns, and authorities may consider to try and coordinate primary vaccination all at the same time. However, the findings may also be historically idiosyncratic to this particular pandemic time. There is a noticeable decrease in mask-related tweets after 2021 as most countries have now lifted masking mandates in public spaces [55].” |
|
14 |
180-183: The inference that "the frequency of tweets increased sharply in year 2020 following the onset of the COVID-19 pandemic, which may reflect the unprecedented impact of the COVID-19 pandemic on public discourse, heightened public awareness and interest in vaccination related topics" could hinge on the question of proportion rather than count. If, for example, both positive and negative tweets increased sharply, and neutral or unclassifiable tweets diminished significantly in proportion, it would fit your claim equally, yet reflect a fundamentally different set of implications for potential policy and intervention. Thus, by focusing only on negative sentiments and combining positive and neutral/unclassifiable into the same category, you obscure such distinction.
|
Thank you for the comment. We apologise for the confusion regarding Figure 4, which is meant to reflect proportion rather than prevalence. We have updated the figure labels accordingly. We have also acknowledged in our discussion of study limitations that, "In our analyses, we also focused solely on negative sentiments and combined positive and neutral/unclassifiable into the same category, which may obscure certain nuances and distinctions. Moreover, some tweets that appear affect neutral or positive may convey negative beliefs." |
|
15 |
87-189: The claim "there appears to be there appears to be some transference of negative thoughts and feelings to influenza vaccination, as can be seen in Topics 3 and 4, which contained skepticism regarding COVID-related policies," again ignores Topic 5, which is also a criticism of government policy. Granted, it is a numerically small category, but nevertheless fits the condition of your claim here.
|
We thank you for highlighting this point. We have edited the statement to include Topic 5.
“Additionally, there appears to be some transference of negative thoughts and feelings to influenza vaccination, as can be seen in Topics 3, 4 and 5 which contained skepticism regarding COVID-related policies, namely the effectiveness of masks to prevent respiratory infections, priority groups to receive the COVID vaccine and concerns with the use of mRNA in vaccines.” |
|
16 |
277-278: "Despite best efforts, we were unable to completely eliminate nonhuman Twitter users (i.e., bots)..." Where do you explain what your "best efforts" were? No such step is explicitly noted in your flowchart selection process. Bots might or might not be caught as "duplicates," depending on how sophisticated the bot program is and which tweet elements you included in your duplicate search. You might want to cite Yuan et al. (2019) to suggest they may represent a small minority (even though various studies of COVID twitter traffic shows everywhere from 2% to two-thirds of such traffic is bot-based). |
We thank the reviewer for the comment as well as the highlighted reference. We have updated the statement with the reference and provided context to our “best efforts”.
Despite best efforts to include tweets by users with real human names and excluding retweets and duplicated tweets, we were unable to completely eliminate nonhuman Twitter users (i.e., bots) posing as genuine users from our study sample. However, these bots have been shown to constitute only a small fraction of users and tweets on the platform [66]. |
|
17 |
218: Yes, corrections and prebunking and nudges show some promise here and there, but it is a bit premature to start setting significant policy choices on such interventions. In research on over 54 million users over five years, Zollo et al. (2017) identified two "well-formed and highly segregated" significant echo chambers interacting on Facebook involving either conspiracy-like posts or scientifically-oriented posts. One of their findings was that "dissenting information online is ignored," such "that debunking information remains confined within the scientific echo chamber and that very few users of the conspiracy echo chamber interact with debunking posts" (p. 9). Reactive debunking (e.g., counter-narrative campaigns, post-hoc corrections, etc.) and particularly preemptive information distortion inoculation strategies e.g., prebunking, preemptive corrections, warnings, critical thinking cautions, discernment of invalid rhetorical tropes, etc.; Bertolotti & Catellani, 2023; Bode & Vraga, 2018; Chan et al., 2017b; Fernández-Garcí & Salgado, 2022; Guan et al., 2021; Lee & Shin, 2021; Lewandowsky et al., 2017; Roozenbeek et al., 2022; van der Linden, Roozenbeek et al., 2020; Vijaykumar et al., 2021) offer significant promise (Ecker et al., 2022), although a meta-analysis of 32 studies of the effect of correcting misinformation found some of these strategies had relatively small effects (Walter & Tukachinsky, 2020). Social and mass media self-regulation, such as with warning or credibility labels, nudges, removal, downranking, and de-platforming demonstrate potential for reducing DIM (Saltz et al., 2021). For example, according to one estimate, de-platforming Trump resulted in at least a temporary 73% reduction in misleading information about election fraud (Dwoskin & Timberg, 2021). In contrast, flagging posts as fake news or lacking credibility appear to elicit some greater engagement, but may not affect beliefs (Moravec et al., 2019). Three studies find evidence that those who are deplatformed for misinformation tend to increase their toxicity elsewhere on other platforms (Ali et al., 2021; Lenti et al., 2022; Mitts et al., 2022). Yet other studies find that many of the self-imposed attempts by social media to control DIM have had limited success (e.g., Quinn et al., 2022).
255-258: Your claim here that "providing accurate information" "unfortunately ... may not be effective" seems to undercut your rationale unless you can provide some suggested direction for future intervention or research other than "a multi-faceted approach."
|
We thank you for the insightful comment and references. We have updated our discussion accordingly.
"While previous studies have found that credibility of correction source matters [45], addressing the issue of vaccine hesitancy is not a straightforward task given the highly polarized nature of vaccine conversations [46]. A study on over 54 million users found that users interacting with conspiracy topics tend to exist in their own well-formed and highly segregated echo chambers, ignoring dissenting information [47]. This suggests that debunking may not be as effective in addressing anti-vaccine sentiments, given the limited interaction of conspiracy users with these debunking posts [47]. However, new strategies such as reactive debunking and particularly preemptive information distortion strategies offers significant promise [48-53]. Alternative measures such as tighter social and mass media regulation can be considered especially in the context of misinformation. Given that anti-vaxxers and partisan actors have been found to be the most influential in shaping antivaccine sentiments on Twitter [46], it is thus important to regulate the content posted by this small but influential group of users. This has proven to demonstrate potential in the reduction of misinformation in other contexts, for example, the de-platforming of Trump resulted in at least a temporary reduction in misleading information about election fraud [54]. Further research is required to develop more effective and comprehensive strategies to combat vaccine hesitancy and misinformation."
|
|
18 |
274-275: Here you admit the limitation that "the tweets collected for analysis were not necessarily from unique users". But elsewhere you make statements as if this were not the case: e.g., 160-162: "Our data 160 revealed that a large majority of the unique users tweeting originated from Europe (n = 161 34,938, 28.1%) and North America", Fig. 1: "Individual users tweets", abstract: "from unique individuals"
|
We thank you for the comment. We apologise for the confusion and we have removed the words “unique” from the descriptions of our dataset. |
|
19 |
Finally, check your reference formatting. There is inconsistency in format. For example, major words in article title are uncapitalized in 46, but capitalized in 47. Sloppy. |
We thank you for the comment. We apologise for the formatting lapses and we have edited the references accordingly. |
Reviewer 3 Report (New Reviewer)
The article "Examining the Negative Sentiments Related to Influenza Vaccination from 2017 to 2022: An Unsupervised Deep Learning Analysis of 261,613 Twitter Posts" deals with a highly relevant topic for global health.
It is essential that public health authorities and public health policy makers for immunization have tools that allow them to deepen their analysis and understanding of issues involving vaccine coverage - the covid-19 pandemic has taught a lot about this.
The article makes clear its objectives and its field of application. Therefore, my recommendations will be towards further improving the article.
Recommendations
The methodology is not clear and transparent, so it needs to be improved.
[1] The authors need to improve the description of the methodology. The methodology must be written in such a way that the experiment can be replicated. I recommend that the authors describe the materials used, justify why they used them and all the steps used (methodological sequence) in the work. Using a flowchart and explaining each part of the flowchart can help with this description.
[2] It is good practice that all Tweets used in the experiment are available in a supplemental file and also the source codes used to process the files. This will improve the transparency of the research and may also help other researchers. Therefore, I recommend that the authors make available the tweets used (they can all be in a .txt file) and the source codes used so that other researchers can test and repeat the experiments. If the authors cannot make the source codes available, they must include pseudocodes (algorithms) in the methodology.
[3] All computational resources and techniques must be justified.
I recommend doing a general revision of the text for the final version.
Author Response
First, we would like to thank the esteemed reviewer for taking the time to review our manuscript.
Comment 1: The authors need to improve the description of the methodology. The methodology must be written in such a way that the experiment can be replicated. I recommend that the authors describe the materials used, justify why they used them and all the steps used (methodological sequence) in the work. Using a flowchart and explaining each part of the flowchart can help with this description.
Reply 1: Thank you for the comments. We have also rephrased our methods section substantially as per advice. The steps have been summarised in a flowchart (Figure 1). As referenced, the sample codes that are used in the methodology are freely available at: https://huggingface.co/dslim/bert-base-NER, https://huggingface.co/siebert/sentiment-roberta-large-english and https://maartengr.github.io/BERTopic. This has now been added to our manuscript.
Comment 2: It is good practice that all Tweets used in the experiment are available in a supplemental file and also the source codes used to process the files. This will improve the transparency of the research and may also help other researchers. Therefore, I recommend that the authors make available the tweets used (they can all be in a .txt file) and the source codes used so that other researchers can test and repeat the experiments. If the authors cannot make the source codes available, they must include pseudocodes (algorithms) in the methodology.
Reply 2: Thank you for the comments and suggestions. We have made the necessary additions. In accordance with Twitter’s terms of use, sample tweets (top 20 tweets) for each topic are shown in the Supplementary Material (Table S1).
Comment 3: All computational resources and techniques must be justified.
Reply 3: Thank you for the comment. We have provided further elaboration in our methods section and also our discussion of study strengths and limitations. In particular, the SieBERT model we utilised for sentiment analysis has demonstrated high accuracy rates of around 93% when tested on various datasets [32], giving us confidence in its ability to accurately identify tone of datasets of text.
Round 2
Reviewer 1 Report (New Reviewer)
The paper has sufficiently improved and is now ready to be published
no comments
Author Response
Thank you for the review!
Reviewer 2 Report (New Reviewer)
Vaccines-2355865-R1-Examining flu vax negative sentiments 2017-2022
It’s a subtle issue concerning Popperian falsification, but technically SieBERT cannot have “proven” its accuracy. Its accuracy can be “demonstrated,” “supported,” “indicated,” or “evidenced,” but none of these can assure any scientific claim to be “proven” (due to the constraints of inferential logic).
65-67: You cite Shah et al. for your claim that “A recent study on public attitudes toward vaccination found a 24% shift toward negative sentiment on social media in wake of the COVID-19 pandemic, 66 citing concerns about safety, efficacy and side effects of vaccination.” However, this is a severe gloss on their results. Overall, pre-COVID negative tweets were 50.19% were negative, and during COVID 46.5% of tweets were negative. Of those shifts of sentiment among individual users, they found that “Overall, in 57,729 unique users, we observed a reduction in the negative sentiments… The overall change in the positive sentiments from period T1 to period T2 was prominent.” They summarized that “the sentiments of about 29.5% (n = 16,976) Twitter users shifted toward positive, and the sentiments of 24% (n = 13,798) users shifted toward negative. The positive shift in sentiments as slightly higher than the negative shift.” This hardly directly supports either your use of the citation or your overall rationale for restricting your analyses to negative sentiment only.
In my initial review I asked whether intercoder reliability was assessed. As one of your own sources argues, “Because of the use of human coders, intercoder reliability needs to be calculated” (Tang et al., 2018).
Figure 4 still has the redundant graph lines, in which the “non-negative” line is uninformative and pragmatically redundant. You note the problem in 325-327, but this does not account for why you retain both trend lines in Fig. 4. You state generically that BERT is a “deep machine learning approach” (97), but what is it “learning” from? It does not appear to be any gold-standard human-coded training set, and instead appears to be based entirely on “unsupervised … pre-training and fine-tuning” (99-100). The limitation of this approach is evident in Topic Category 3, where every single one of the 20 exemplary tweets has “asthma” in it, which reflects a rather narrow topic tributary.
322-324: There is additional rationale for examining negative sentiments, but none of the concerns I raised about the differential dynamics of distinct negative sentiments (e.g., sadness, anger, fear, and disgust) are noted. Negative sentiments is a very, very blunt construct and there is plenty of sentiment analysis that implies the importance of differentiating types of negative sentiment (as well as differentiating negative sentiments from both “neutral” and “positive” sentiments, rather than just “non-negative” message content).
The importance of this seems substantial. For example:
· Topic 1, exemplar #10: “If the vaccine stopped someone from contracting it and spreading Covid I think you would have close to 100% mandate compliance. Unfortunately we re not there yet so to mandate it and fire people for somthing that basically works like a flu vaccine is sad.” Clearly the keyword “sad” is likely a cue for coding this as negative, but while it implies categorizing it as “sadness,” it probably illustrates anger. Research has demonstrated that sadness and anger tend to diffuse differentially.
· Topic 1, exemplar #20: “Also, unvaccinated people can prolong the pandemic since they can overwhelm hospitals requiring vaccinated people to social distance and wear masks. Taking away the freedoms of everyone else. I don't think the MMR or Flu Vaccine should be mandatory but Covid vaccine should be.” This is an ambivalent text, in which a negative/con (“I don't think the MMR or Flu Vaccine should be mandatory”) is nuanced with a positive/pro (“but Covid vaccine should be”). It qualifies as negative for flu, but not for COVID-19, or, for one vaccine but not another vaccine.
· Topic 1, exemplar #16: “So does this mean they have never agreed to have any vaccines, ever? I ve never heard of people rejecting the flu vaccine or say, typhoid vaccine if they re travelling to affected counties. Who is delivering this message to ordinary families?” This sounds negative if it is assumed that curiosity, uncertainty or surprise, but hardly qualifies definitively as negative.
· Topic 5, exemplar #16: “The flu vaccine uses 50-year-old technology. We should replace it with an mRNA vaccine, which will be far more effective.” This seems distinctly positive in net affect, as it is hardly a direct criticism of the flu vaccine so much as to say there may be something better. Topic 2 #1 seems similarly neutral to positive in sentiment.
· Topic 2, #5: “He's recently made negative comments about the flu vaccine.” This is not a negative tweet—it is a description of something negative that someone else said, and thus, should be considered neutral. Similarly, Topic 2, #6: “Heard him say that at the rally, but heard a radio interview from last week where he said he didn't. Never takes flu vaccine either.” Seems more neutral-descriptive than negative in affect. I also cannot formulate a strong reason to consider Topic 2, #7 as negative in sentiment: “I m not sure Bill believes the covid vaccine works given his previous advocacy agaisnt the flu vaccine.”
· It’s not obvious how Topic 2, #17: “Wrong. Problem is covid vaccines were, out of necessity, rushed out, with minimal testing. Trump gets credit for working with Pharma to meet the timeline, but any credible medical researcher knows the risks involved. Standard flu vaccine went through the complete FDA protocol.” Differs from Topic 5, #4: “Also, there's a big difference between getting the flu jab, which has been through rigorous trials, testing, peer-review, and risks now known to us, than there is when taking this new """"RNA-treatment"""", which its risks are completely unknown to us long-term.”
All sentiment and topic systems make these kinds of mistakes, but if no training occurred through a stage of human validation and intercoder reliability assessment, it is difficult for a reader to ascertain the value of the data or your conclusions.
335-339: Again, you still do not explain what your “best efforts” were to exclude bots. I’m guessing that this is implied by your statement in 102-104: “Named Entity Recognition was then used to identify individual users, focusing on 102 location, organizations, person, and miscellaneous entities [32]. To accomplish this, actual 103 human names were used to identify the Twitter account associated with each post.” If so, then you need to (1) indicate that this is your approach and (2) cite evidence and sources indicating the efficacy of this approach to identifying/extracting/excluding bots.
Given that identifying bots is a major taproot in big data analytics, it is essential that some explanation be provided. If bots represent a major percentage of your data, then it seriously undermines almost all other claims made. In your Fig. 1, you move from the “Duplicate tweets” removed step to “Unique tweets” to “Individual users tweets”, with no explanation of how these exclusions occurred other than removing “duplicate tweets,” which is a crude way of controlling for bots.
By some estimates, bots represent 8% to 35% of all Twitter traffic (Hindman & Barash, 2018; Huang & Carley, 2020; Moffitt et al., 2021; Orabi et al., 2020; Zhang et al., 2023) and play a substantial role (17-29%) in (mis)information spread (e.g., Jamison, Broniatowski, Dredze et al., 2020; Huang & Carley, 2020; Memon & Carley, 2020; Moffitt et al., 2021; Weng & Lin, 2022; Zhen et al., 2023). In one study by Pew Research (Wojcik et al., 2018), “of all tweeted links to popular websites, 66% are shared by accounts with characteristics common among automated ‘bots,’ rather than human users” and “among popular news and current event websites, 66% of tweeted links are made by suspected bots” (p. 4). Bots represent from about 2-5% (Caldarelli et al., 2021; Gruzd & Mai, 2020) to a fifth to two-thirds (Ayers et al., 2021; Chong & Park, 2021; Himelein-Wachowiak et al., 2021; Memon & Carley, 2020; Sacco et al., 2021; Scannell et al., 2021; Xu & Sasahara, 2021; Zhang et al., 2023) of COVID-19 related content in collected corpuses. A study of COVID-19 fact-checked tweets estimated that “more than 15% percent of the data on Twitter about misinformation is cloned content from existing tweets to increase the impact of the message, and the half of it belongs to tweets that have 10 or more clones” (Villar-Rodríguez et al., 2022, p. 250). A study of Twitter data regarding the Wuhan lab leak conspiracy theory estimated that 29% were likely bots, which appeared to play “a bridge role in diffusion” of the topic (Weng & Lin, 2022). In another study Twitter accounts with higher bot scores posted approximately “27 times more about COVID-19 than those with the lowest bot scores” (Ferrara, 2020; see also: Broniatowski et al., 2018). Another study of tweets about the pandemic found that social bot tweets were retweeted more than human account tweets, indicating that “content about public health published by social bot accounts is more able to attract the attention of social network users and has won the recognition and attention of a lot of users” (Zhang et al., 2023, p. 14; cf., Cai et al., 2023; Zhen et al., 2023).
So one argument is that if bots are in the digital circulatory system, their sentiment still counts because their sentiment is still what people are (potentially) exposed to. However, flu-vaccine bot-based tweets may or may not diffuse in the same way as real tweets. Are people more or less likely to encounter, read, engage and/or respond to bot-based tweets compared to authentic tweets? Are the bots more negative than the population of tweets? How do we know if you did not analyze the bot-based tweets? If your only approach to bot management was to exclude redundant tweets then this technique needs to be defended with research that shows it is reasonably effective. It is not clear to what extent AI is transforming the variability of bot-based tweets and texts so that mere redundancy might not be a sufficient guardrail.
As one of your own sources cited points out, “studies of EID [emerging infectious diseases] communication on social media are not very theoretical in general” (Tang et al., 2018). The word theory appears nowhere in this article, and the term “theoretical” appears incidentally only once in text, in response to a concern I raised in my preliminary review. In my initial review I was concerned about the generalizability of your typology, and I still find the state of theory sadly anemic in such work.
Cervi et al., 2021, you are missing its doi: https://doi.org/10.3390/socsci10080294
Author Response
To begin, we would like to thank the reviewer for taking the time and effort to review our manuscript and provide constructive feedback for improvement. Our replies are enclosed below and changes in the manuscript were highlighted in yellow for easy identification.
|
S/N |
Comment |
Reply |
|
1 |
It’s a subtle issue concerning Popperian falsification, but technically SieBERT cannot have “proven” its accuracy. Its accuracy can be “demonstrated,” “supported,” “indicated,” or “evidenced,” but none of these can assure any scientific claim to be “proven” (due to the constraints of inferential logic). |
Thank you for the comment. We have made the necessary amendment. The word ‘demonstrated’ is now used in place of ‘proven’. |
|
2 |
65-67: You cite Shah et al. for your claim that “A recent study on public attitudes toward vaccination found a 24% shift toward negative sentiment on social media in wake of the COVID-19 pandemic, 66 citing concerns about safety, efficacy and side effects of vaccination.” However, this is a severe gloss on their results. Overall, pre-COVID negative tweets were 50.19% were negative, and during COVID 46.5% of tweets were negative. Of those shifts of sentiment among individual users, they found that “Overall, in 57,729 unique users, we observed a reduction in the negative sentiments… The overall change in the positive sentiments from period T1 to period T2 was prominent.” They summarized that “the sentiments of about 29.5% (n = 16,976) Twitter users shifted toward positive, and the sentiments of 24% (n = 13,798) users shifted toward negative. The positive shift in sentiments as slightly higher than the negative shift.” This hardly directly supports either your use of the citation or your overall rationale for restricting your analyses to negative sentiment only.
|
Thank you for your comment. We apologise for the misquotation and we have revised and updated the paragraph accordingly.
“In the recent COVID-19 pandemic, social media has contributed to the proliferation of misinformation [18], undermining public trust and contributing to vaccine hesitancy amongst the public [19]. Furthermore, it is concerning that influential figures, particularly right wing politicians, utilise platforms such as Twitter for political discourse to shape public perception on pandemics [20,21]. With more than 200 million daily active users [22], Twitter is a powerful platform for the rapid dissemination of information and expression of emotions. Given the wealth of information on Twitter, it has served to be useful in assessing public views on health issues [23,24]. Sentiment analysis of tweets can thus help to better understand public perception toward vaccination. A recent study using sentiment analysis has demonstrated that tweet sentiment score may help to predict vaccination rates, with negative sentiment being predictive of lower vaccination rates [25]. Given that misinformation tended to be emotionally driven, demonstrate more negative sentiments such as fear and anger [26,27], and that anti-vaccine twitter users were more likely to spread unreliable information [28], we chose to focus on negative sentiment analysis to provide insight into reasons for resistance to vaccination and to identify potential sources of misinformation. “
|
|
|
In my initial review I asked whether intercoder reliability was assessed. As one of your own sources argues, “Because of the use of human coders, intercoder reliability needs to be calculated” (Tang et al., 2018).
|
Thank you for your comment. We acknowledge the limitations of our methods and we have amended the limitations accordingly.
“In our study, we utilized sentiment analysis based on machine learning which comes with its own strengths and limitations. While it allows for the study of a large number otherwise not possible with human coding, we acknowledge that it is not as accurate as human coding. Additionally, as the coding was performed in team through an iterative matter via discussion and consensus, we were not able to calculate intercoder reliability.”
|
|
|
Figure 4 still has the redundant graph lines, in which the “non-negative” line is uninformative and pragmatically redundant. You note the problem in 325-327, but this does not account for why you retain both trend lines in Fig. 4.
|
Thank you for your comment. We have updated the graph to include only the negative sentiment line.
|
|
|
You state generically that BERT is a “deep machine learning approach” (97), but what is it “learning” from? It does not appear to be any gold-standard human-coded training set, and instead appears to be based entirely on “unsupervised … pre-training and fine-tuning” (99-100). The limitation of this approach is evident in Topic Category 3, where every single one of the 20 exemplary tweets has “asthma” in it, which reflects a rather narrow topic tributary.
|
Thank you for the comment. We apologise for the confusion. We have added a statement to clarify the source of learning for BERT.
“BERT utilizes unsupervised masked language model (MLM) and unsupervised next sentence prediction (NSP) for text deep pre-training and fine-tuning using publicly available two-class sentiment datasets, as opposed to the traditional bag-of-words model [31].” |
|
|
322-324: There is additional rationale for examining negative sentiments, but none of the concerns I raised about the differential dynamics of distinct negative sentiments (e.g., sadness, anger, fear, and disgust) are noted. Negative sentiments is a very, very blunt construct and there is plenty of sentiment analysis that implies the importance of differentiating types of negative sentiment (as well as differentiating negative sentiments from both “neutral” and “positive” sentiments, rather than just “non-negative” message content). The importance of this seems substantial. For example: · Topic 1, exemplar #10: “If the vaccine stopped someone from contracting it and spreading Covid I think you would have close to 100% mandate compliance. Unfortunately we re not there yet so to mandate it and fire people for somthing that basically works like a flu vaccine is sad.” Clearly the keyword “sad” is likely a cue for coding this as negative, but while it implies categorizing it as “sadness,” it probably illustrates anger. Research has demonstrated that sadness and anger tend to diffuse differentially. · Topic 1, exemplar #20: “Also, unvaccinated people can prolong the pandemic since they can overwhelm hospitals requiring vaccinated people to social distance and wear masks. Taking away the freedoms of everyone else. I don't think the MMR or Flu Vaccine should be mandatory but Covid vaccine should be.” This is an ambivalent text, in which a negative/con (“I don't think the MMR or Flu Vaccine should be mandatory”) is nuanced with a positive/pro (“but Covid vaccine should be”). It qualifies as negative for flu, but not for COVID-19, or, for one vaccine but not another vaccine. · Topic 1, exemplar #16: “So does this mean they have never agreed to have any vaccines, ever? I ve never heard of people rejecting the flu vaccine or say, typhoid vaccine if they re travelling to affected counties. Who is delivering this message to ordinary families?” This sounds negative if it is assumed that curiosity, uncertainty or surprise, but hardly qualifies definitively as negative. · Topic 5, exemplar #16: “The flu vaccine uses 50-year-old technology. We should replace it with an mRNA vaccine, which will be far more effective.” This seems distinctly positive in net affect, as it is hardly a direct criticism of the flu vaccine so much as to say there may be something better. Topic 2 #1 seems similarly neutral to positive in sentiment. · Topic 2, #5: “He's recently made negative comments about the flu vaccine.” This is not a negative tweet—it is a description of something negative that someone else said, and thus, should be considered neutral. Similarly, Topic 2, #6: “Heard him say that at the rally, but heard a radio interview from last week where he said he didn't. Never takes flu vaccine either.” Seems more neutral-descriptive than negative in affect. I also cannot formulate a strong reason to consider Topic 2, #7 as negative in sentiment: “I m not sure Bill believes the covid vaccine works given his previous advocacy agaisnt the flu vaccine.” · It’s not obvious how Topic 2, #17: “Wrong. Problem is covid vaccines were, out of necessity, rushed out, with minimal testing. Trump gets credit for working with Pharma to meet the timeline, but any credible medical researcher knows the risks involved. Standard flu vaccine went through the complete FDA protocol.” Differs from Topic 5, #4: “Also, there's a big difference between getting the flu jab, which has been through rigorous trials, testing, peer-review, and risks now known to us, than there is when taking this new """"RNA-treatment"""", which its risks are completely unknown to us long-term.” All sentiment and topic systems make these kinds of mistakes, but if no training occurred through a stage of human validation and intercoder reliability assessment, it is difficult for a reader to ascertain the value of the data or your conclusions.
|
Thank you for the comment. We agree and we acknowledge the differential diffusion of emotions and the limitations of our methods. Unfortunately, there are several inherent limitations to using unsupervised machine learning to analyse a large corpus of tweets.
“In our study, we utilized sentiment analysis based on machine learning which comes with its own strengths and limitations. While it allows for the study of a large number otherwise not possible with human coding, we acknowledge that it is not as accurate as the latter, especially with unsupervised machine learning. Additionally, as the coding was performed in team through an iterative matter via discussion and consensus, we were not able to calculate intercoder reliability. Furthermore, the pre-trained SieBERT model we utilized for sentiment analysis is only capable of classifying tweets into negative versus positive sentiment [32], thus we were unable to differentiate amongst the different types of negative sentiments, e.g. fear, anger and shame. In our analyses, we also focused solely on negative sentiments and combined positive and neutral/unclassifiable into the same category, which may obscure certain nuances and distinctions. Moreover, some tweets that appear affect neutral or positive may convey negative beliefs.” |
|
|
335-339: Again, you still do not explain what your “best efforts” were to exclude bots. I’m guessing that this is implied by your statement in 102-104: “Named Entity Recognition was then used to identify individual users, focusing on 102 location, organizations, person, and miscellaneous entities [32]. To accomplish this, actual 103 human names were used to identify the Twitter account associated with each post.” If so, then you need to (1) indicate that this is your approach and (2) cite evidence and sources indicating the efficacy of this approach to identifying/extracting/excluding bots. Given that identifying bots is a major taproot in big data analytics, it is essential that some explanation be provided. If bots represent a major percentage of your data, then it seriously undermines almost all other claims made. In your Fig. 1, you move from the “Duplicate tweets” removed step to “Unique tweets” to “Individual users tweets”, with no explanation of how these exclusions occurred other than removing “duplicate tweets,” which is a crude way of controlling for bots. By some estimates, bots represent 8% to 35% of all Twitter traffic (Hindman & Barash, 2018; Huang & Carley, 2020; Moffitt et al., 2021; Orabi et al., 2020; Zhang et al., 2023) and play a substantial role (17-29%) in (mis)information spread (e.g., Jamison, Broniatowski, Dredze et al., 2020; Huang & Carley, 2020; Memon & Carley, 2020; Moffitt et al., 2021; Weng & Lin, 2022; Zhen et al., 2023). In one study by Pew Research (Wojcik et al., 2018), “of all tweeted links to popular websites, 66% are shared by accounts with characteristics common among automated ‘bots,’ rather than human users” and “among popular news and current event websites, 66% of tweeted links are made by suspected bots” (p. 4). Bots represent from about 2-5% (Caldarelli et al., 2021; Gruzd & Mai, 2020) to a fifth to two-thirds (Ayers et al., 2021; Chong & Park, 2021; Himelein-Wachowiak et al., 2021; Memon & Carley, 2020; Sacco et al., 2021; Scannell et al., 2021; Xu & Sasahara, 2021; Zhang et al., 2023) of COVID-19 related content in collected corpuses. A study of COVID-19 fact-checked tweets estimated that “more than 15% percent of the data on Twitter about misinformation is cloned content from existing tweets to increase the impact of the message, and the half of it belongs to tweets that have 10 or more clones” (Villar-Rodríguez et al., 2022, p. 250). A study of Twitter data regarding the Wuhan lab leak conspiracy theory estimated that 29% were likely bots, which appeared to play “a bridge role in diffusion” of the topic (Weng & Lin, 2022). In another study Twitter accounts with higher bot scores posted approximately “27 times more about COVID-19 than those with the lowest bot scores” (Ferrara, 2020; see also: Broniatowski et al., 2018). Another study of tweets about the pandemic found that social bot tweets were retweeted more than human account tweets, indicating that “content about public health published by social bot accounts is more able to attract the attention of social network users and has won the recognition and attention of a lot of users” (Zhang et al., 2023, p. 14; cf., Cai et al., 2023; Zhen et al., 2023). So one argument is that if bots are in the digital circulatory system, their sentiment still counts because their sentiment is still what people are (potentially) exposed to. However, flu-vaccine bot-based tweets may or may not diffuse in the same way as real tweets. Are people more or less likely to encounter, read, engage and/or respond to bot-based tweets compared to authentic tweets? Are the bots more negative than the population of tweets? How do we know if you did not analyze the bot-based tweets? If your only approach to bot management was to exclude redundant tweets then this technique needs to be defended with research that shows it is reasonably effective. It is not clear to what extent AI is transforming the variability of bot-based tweets and texts so that mere redundancy might not be a sufficient guardrail.
|
We thank you for the insightful comment and for the helpful references provided. We acknowledge the limitations of our method of excluding the bots as well as the arising implications. We have updated our limitations accordingly to reflect this.
“Despite our efforts to include tweets by users with real human names and excluding retweets and duplicated tweets, we were unable to completely eliminate nonhuman Twitter users (i.e., bots) posing as genuine users from our study sample. Given that it is unclear to what extent artificial intelligence is transforming the variability of bot-based tweets, we acknowledge that this measure alone may not be sufficient guardrail. A recent study on social bot’s involvement in the COVID-19 vaccine discussions found that 8.87% of the users identified were bots, and these bots contributed to 11% of the tweets [66]. These bots may have been designed to manipulate public opinion and disseminate false information, which could potentially impact our analyses.”
|
|
|
As one of your own sources cited points out, “studies of EID [emerging infectious diseases] communication on social media are not very theoretical in general” (Tang et al., 2018). The word theory appears nowhere in this article, and the term “theoretical” appears incidentally only once in text, in response to a concern I raised in my preliminary review. In my initial review I was concerned about the generalizability of your typology, and I still find the state of theory sadly anemic in such work. |
We thank you for the comments. We agree and we acknowledge the limitations of our study. We have added the following point to our limitations accordingly.
“Firstly, we aimed to explore the phenomenon of negative sentiments on Twitter in an exploratory manner. As such, we did not have any pre-established theoretical framework guiding our research. While the absence of a theoretical framework allowed us to approach the topic with an open mind, and explore various aspects, patterns, and relationships that emerged during the study, we acknowledge the lack of theoretical grounding for our study and the potential lack of generalizability of the typology uncovered herein.” |
|
|
Cervi et al., 2021, you are missing its doi: https://doi.org/10.3390/socsci10080294 |
Thank you for pointing this out. We apologise and have added the corresponding doi to the reference. |
This manuscript is a resubmission of an earlier submission. The following is a list of the peer review reports and author responses from that submission.
Round 1
Reviewer 1 Report
This manuscript deals with an interesting topic, but the methodology needs some amendments and the results need to be described in more detail.
In the introduction section, the reported influenza vaccination rates for the different countries need to be updated and should include a range of rates covering the whole study period. Accordingly, the references need to be updated too.
In the results section, the temporal trends need more detail in numbers and could be improved by analyzing the trend for each type/topic of criticism against the influenza vaccine as stated in Table 1.
The discussion describes issues on vaccine hesitancy in general but could focus more on the association of COVID-19 pandemic with the dropping rates of vaccination coverage.
The authors do not describe in detail the countries of origin of these tweets and how the study has addressed the major methodological issues that are stated in the limitations. Although the conclusion can be interpreted in the context of the present situation, addressing these methodological issues is important for augmenting the reliability of the results.
Reviewer 2 Report
This paper, “Examining the Negative Sentiments Related to Influenza Vaccination from 2017 to 2022: An Unsupervised Deep Learning 3 Analysis of 261,613 Twitter Posts” describes themes based on one quarter million tweets.
The term unique tweets need to be tempered with the comment that this ‘unique tweets’ likely does not match unique senders. Frequent tweets are an important component in maintaining a following and therefore one limitation of this study is a lack of understanding about the representativeness of these findings. It could be from a very active few or could be more widespread. This point should be included in the discussion.
Another limitation of this study is the lack of context. Country based sediments are quite varied and not being able to place these in context severely limits the applicability of these findings. One of the statements at the end alluded to knowing the place of the tweet “The analysis was based on Twitter posts 205 (with users chiefly from North America)” in the limitation section. Adding this would be important as it provides context.
The graph showing the trend is interesting but it is not clear why certain topics were lumped together versus showing 5 lines, one for each theme.
Overall, this seems limited in our understanding of comments w/r to public opinion on influenza given 1) no indication of country – lacking context, 2) no indication of unique people posting, 3) no indication of the demographics of posters, 4) very broad themes.
What is missing is tweets that connect COVID perceptions with influenza—the positive and negative. That was not the focus of this paper but suggested that COVID perceptions spilled over and influenced Influenza perceptions. Data to support that concept would be interesting.
In addition, there was not a temporal description of the tweets. In other words, were the same themes proportionally indicated pre COVID (before 2020)? Along the same lines, what was the volume of tweets overall, since Fig 2 only shows negative. Was there an upsurge in tweeting on this topic? This should also be described.